# An evidence-based approach to assessing the effectiveness of training regimen on athlete performance: Youth soccer as a case study

**Cam M. K. Rechenmacher[1], Michael Keating[2], James D. Nichols[3,4], Jonathan M. Nichols[4]** *

**1** University of North Carolina, Chapel Hill, NC, United States of America, **2** Captain Elite, Soccer Research and Training Organization, Oak Ridge, NC, United States of America, **3** University of Florida, Gainesville, FL, United States of America, **4** Developmental Sports Analytics, Crofton, MD, United States of America

* jamesdnichols2@gmail.com

**Data Availability Statement:** All relevant data are within Supplemental Files S2 and S3

## Abstract

Athletic performance data are modeled in an effort to better understand the relationship between both hours spent training and a measurement of "commitment" to that training, and improvements in performance. Both increased training time and greater commitment were predicted to produce larger increases in performance improvement, and commitment was predicted to be the more important determinant of improvement. The performance of 108 soccer players (ages 9–18) was quantified over a 10-week training program. Hours spent training ranged from 16 to 90 during the course of the study, while commitment scores ranged from 0.55 to 2.00, based on a scale from 0.00 to 2.40. A model selection approach was used to discriminate among models specifying relationships between training hours and improvement, and commitment and improvement. Despite considerable variability in the data, results provided strong evidence for an increase in performance improvement with both training hours and commitment score. The best models for hours and commitment were directly compared by computing an evidence ratio of 5799, indicating much stronger evidence favoring the model based on commitment. Results of analyses such as these go beyond anecdotal experience in an effort to establish a formal evidentiary basis for athletic training programs.

## Introduction

Coaches, trainers and athletes have searched for the optimal formula for skill acquisition for as long as sports have existed, simply because of the competitive advantages conferred on athletes who can develop abilities faster, or to a greater extent, than others [1, 2]. Historically, the selection of training methods has been guided primarily by anecdotal experience rather than by empirical evidence on effectiveness [3, 4]. There have been some efforts to develop evidence for the effects of training on sports and other endeavors [1, 5, 6]. However, based on varied experiences of the authors with elite club, high school, Division 1 collegiate, and professional athletics, it appears that athletes and their mentors seldom use strong empirical evidence to

**Funding:** The authors received no specific funding for this work.

**Competing interests:** The authors have declared that no competing interests exist.

guide their training decisions. Instead, decisions about methods to promote skill development tend to be based on anecdotal experiences about what methods have, and have not, "worked". In some cases, opinions of coaches and trainers can seem to converge on a conventional wisdom. However, the recent revolution in analytics provides strong evidence of the fallibility of such wisdom [3, 7].

Science is based on the key step of comparing hypothesis-based predictions against observations. When hypotheses are consistently good predictors, they inspire confidence and are used to make decisions. This consistency of predictions and observations is referred to as evidence. Greater reliance on evidence-based decisions has been recommended in such diverse fields as dentistry [8], medicine [9], physical education [4] and conservation [10]. In addition to decisions being "better" (more likely to attain objectives), evidence-based decisions are transparent, defensible, objective, and scientific, all desirable attributes of decisions that are subject to scrutiny.

In this paper, the initial steps of evidence-based learning are applied to a case study involving an online program of soccer exercises developed and administered by one of the authors (M.K.). The soccer exercises were unsupervised and administered remotely via video, such that different athletes enrolled in the program expended different amounts of time on the exercises (hours) and obtained different scores for the quality of training and task completion (commitment). Tasks included not only completing training exercises, but also watching motivational videos, updating skill scores, and honoring commitments to training goals. The utility of these two independent variables, hours and commitment, was assessed by the athletes' performance during tests designed to measure their skill levels. Specifically, performance metrics were obtained by self-administered tests near the beginning of the remote 10-week training program and then at the end of this program, permitting the computation of an "improvement" metric for each athlete.

An information-theoretic model selection approach with generalized linear models was used to formally address three questions:

1. Were more training hours associated with larger improvement metrics?

2. Were larger commitment scores associated with larger improvement metrics?

3. Was one of these variables a better predictor of improvement than the other?

Multi-model inference was used to estimate the relationship between skill improvement and these two variables, hours and commitment. These questions were motivated by hypotheses about the adequacy of practice alone [2, 11] as a predictor of skill development, versus the need to also include intrinsic psychological qualities of the athletes [12].

## Hypotheses and predictions

The training program entailed a series of exercises designed to improve the athlete's abilities in several soccer skills important to match play. Data on amount of time spent in program training, commitment and skill level were obtained periodically (e.g., weekly). The focus for this study was a 10-week period. This was the standard length of a training course in this program, selected based on the reported time required for humans to develop new habitual behavior [13]. The training program was flexible such that each athlete decided how much time to spend training, depending on such variables as personal dedication and amount of available time. The central questions are: do time spent in program training and individual commitment affect skill improvement. The basic predictions were that athletes who expended more time training and who exhibited more commitment would show a greater improvement in the

targeted skill. Different plausible hypotheses about the nature and form of these relationships were considered.

The basic hypothesis for the training time analysis was that improvement would be greater for athletes who expended more time training [2, 11]. This hypothesis was incorporated into a model in which improvement increased linearly with training time. A null hypothesis model included no effect of training time on improvement. Development of any skill using any method will vary across individual athletes; hence the need for statistical inference. This analysis was based on a relatively homogeneous group of athletes; soccer players aged 9–18 years who competed at varying levels (recreational through elite). Variation was still expected among athletes in this group with respect to initial skill level. This variation was a primary reason for focusing on improvement in skill over the period of training, rather than on absolute skill level attained (see *Analytic Methods*). In addition, a general model was developed to incorporate the hypothesis that improvement would require more time for athletes beginning the program at a high level of skill. The premise was that athletes starting out at lower skill levels can increase proficiency rapidly. Another general model did not incorporate this hypothesis about starting skill level, but included a term that permitted an inflection in the relationship between improvement and time spent training. Specifically, a leveling off in the training-improvement relationship was predicted to occur at higher levels of training time.

The same basic model set was used for the independent variable, commitment score, as well. Commitment was viewed as a characteristic of each individual athlete and was measured by accumulated activities (see *Training Methods and Metrics*). One model incorporated the hypothesis that improvement was a linear function of commitment (athletes with greater commitment were expected to exhibit greater improvement). In contrast, a null model included no relationship between commitment and improvement. Another model incorporated the hypothesis that initial proficiency influenced the change in improvement with commitment. A final model allowed an inflection, such that the rate of improvement with commitment was lower at high levels of commitment.

## Methods

### Training methods and metrics

**Training program.** This training program utilized a web-based player development platform to which players gained access for 10 weeks when they signed up to participate. Athletes in the program were able to access their performance data, but could not edit the data. For this study, $N_A$ = 108 athletes who supplied usable data from the Spring 2020 training program. Data were not included from individuals who began the training but did not complete the full program for any reason (e.g., injury). Players were able to sign up for one of three skill development tracks for this 10-week period. These tracks each focused on a unique set of skills, the first, dribbling; the second, first touch and passing; and the third, striking.

### Ethics

Approval from an ethics committee was not sought because:

1. the data were collected under the auspices of "Captain Elite, Soccer Research and Training Organization", which has no institutional ethics committee;

2. the focal training segment was one component of an ongoing training program and not designed as a specific research study;

3. this study is retrospective and was not anticipated when the data collection took place;

4. all participants and their parents (in the case of minors) signed consent forms that their data could be used for research purposes;

5. all data were anonymized and analyzed anonymously. Training and performance data had no linkages to individual identification information.

Consent was informed and documented via signature. All athletes and parents/guardians signed a waiver/release at every enrollment that included the following statements:

"I understand that the course involves sharing photos, videos and data of Participant during Participant's training in order to help assess Participant's level, to share with others in the course and to use for anonymous research purposes. I consent for the Company to use these photos, videos and data for these purposes. If I choose not to participate in sharing data, photos and/or videos of Participant's progress, I understand that it is my obligation not to enroll in any of the Company's online training courses. I HAVE CAREFULLY READ THIS RELEASE AND AGREE ON BEHALF OF MY MINOR CHILD."

The collector (M.K.) of the data used in this study is developer and director of a program focused on the training of athletes. This training requires periodic assessment tests to evaluate the rate of development of the athletes (see **Skill improvement**). These tests are periodically evaluated and results posted so that athletes can see their progress and relative standing. Athletes (and their parents/guardians) who participate in this program sign the above waiver/release specifying that they are aware of the evaluation process and the potential uses of their data, and they are the ones who submit test results and associated video evidence.

## Commitment

The program required players to submit a weekly log by midnight every Sunday as a self-report of the work they completed in the training program that week. This log asked players:

1. to record the total number of hours they trained in the training program in the specified week;

2. to specify whether they watched a professional soccer game in the focal week and, If yes, what teams competed;

3. to specify whether they completed each of the tasks required of them for that week (players had to check a box next to each individual task completed for the week).

Each player's commitment score was based on her/his completion of each weekly task, in addition to whether or not s/he met or exceeded the pre-set goal in weekly hours trained. Additionally, players were given the opportunity to earn extra "points" for their commitment scores by completing optional tasks.

Overall commitment scores were computed as the sum of two components, one based on required tasks and the other based on optional tasks. The "required" component was computed as the fraction of required tasks that was completed; i.e., by dividing the number of points accrued by the maximum number of points for a player completing 100% of required tasks. The required component score thus took values between 0 (no required tasks completed) and 1.00 (all required tasks completed). Optional tasks were available as well, with up to a 1.40 score for an athlete scoring maximum optional points. Thus, commitment scores could range from 0 to 2.40 for each player. The commitment metric was intended to go beyond time expended on training, as it incorporated information that reflected an athlete's commitment to completing all tasks required of them, as well as some related tasks that were not required.

## Hours trained

In the weekly log, players self-reported the number of hours they trained on drills in their specific development tracks. Hours trained were computed for each player by summing the number of training hours reported across the 10-week period.

## Skill improvement

The training platform included videos of 10–12 soccer drills that players watched and attempted to replicate on their own. A minimum of three times per 10-week program (beginning, mid-point, and end of the 10 weeks), and a maximum of once per week during the 10 weeks, players were required to self-report their scores on a designated three of these video-based exercises to measure their improvement. Each time players submitted new scores on these 3 drills, they were required to submit videos of themselves completing each drill as evidence that they accomplished the reported scores. Players were encouraged to update their skill scores more frequently, and each time they did so (in excess of the 3 required tests), they were rewarded by gaining more "commitment points".

Each development track had a different scoring system for measurement of skill improvement. Players measured their skill scores in the dribbling track based on how long it took them to perform each of the 3 designated drills. For the first touch and passing track, they recorded how many correct repetitions they could complete in 30 seconds, and for the other 2 drills they recorded how many consecutive, correctly-executed repetitions they could achieve before making a mistake. For the striking track, players measured all 3 of the drills as the distance from the goal at which they could correctly complete each of the striking techniques. Each striking drill was scored separately as an average of the maximum left-footed and right-footed distances the player achieved. A new distance for either foot on any one of the drills could be achieved only by completing 5 correct strikes in a row with that technique.

A player's scores from each of the separate development tracks were combined into one overall score using a "Skill Stage" scoring system reflecting objective standards of skill achievement. Scores ranged between 0 and 4 for each development track (0–1.99 = fundamental youth skill, 2–2.99 = elite youth skill, 3–3.99 = collegiate skill, 4+ = professional skill). For this study, skill improvement was measured as a change in average Skill Stage score across each of the 3 designated drills from the beginning of the 10 weeks until the end of the 10 weeks.

## Analytic methods

Two sets of analyses were conducted for the 2 different independent variables. The first analysis was based on time spent training over the 10-week training period. The participating athletes showed substantial variation in this metric, ranging from about 16 hours to 90 hours. The second set of analyses was based on the composite metric, commitment, that was computed using a point system, where players were awarded a specific number of points according to which of the required and optional tasks they completed throughout the program (see above).

## Training time

Competing models of skill improvement as a function of time spent training were developed and then fit to the data. Model selection was used to assess the level of support for each model. For each of the $i = 1...N_A$ athletes, the primary data used in the analysis were:

$y_{it}$ = skill level of athlete $i$ at beginning of week $t$ as assessed by skill test,

$x_{it}$ = time spent training by athlete $i$ between weeks $t$ and $t+1$.

The focal response variable of the analysis was a statistic reflecting improvement between the initiation of the training program, week $t_0$, and some endpoint assessment time (in this case 10 weeks later), $t_T$:

$$\Delta_i = y_{it_T} - y_{it_0}.$$

The first independent variable of primary interest was accumulated training time between the beginning and end of the training period (denote the length of the training period as $\Delta t = 10$ weeks):

$$x_{i\Delta t} = \sum_{t=t_0}^{t_T} x_{it}.$$

One other independent variable was used in one of the general models for training time, $y_{it_1}$. This variable provided an assessment of skill level at the beginning of the training and assessment period. We did not use $y_{it_0}$ as an assessment statistic, because this would induce a sampling covariance between this statistic and our response variable, $\Delta_i$. Use of $y_{it_1}$ as an independent variable represented an effort to investigate the hypothesis that initial skill level might influence the increases in skill level expected to accompany increases in training time.

The basic approach of this investigation was to develop 4 models that represented 4 competing hypotheses about the relationship between training time and skill improvement (described above). In all models a normal distribution was assumed with mean $\mu$ and variance $\sigma^2$ for the response variable, i.e.,

$$\Delta_i \sim N(\mu, \sigma^2).$$

The models differed only in the structure imposed on the mean, i.e., only in the key determinants of skill improvement. The simplest such model is the null model *1* under which training time beyond 16 hours (the minimum training time among all participants) did not influence skill improvement:

$$\mu_1 = \beta_0.$$

Note that model *1* can be viewed as null with respect to the effect of accumulated hours of training on proficiency. All athletes trained for at least 16 hours during the 10-week program, so $\mu_1$ reflects a baseline level of proficiency that is likely influenced by the first 16 hours of training.

Model *2* was the basic linear model postulating an increase in proficiency with training (prediction: $\hat{\beta}_1 > 0$, where the hat denotes an estimate):

$$\mu_2 = \beta_0 + \beta_1 x_{i\Delta t}.$$

Note that no data were available on the effects of the first few hours of training, and rapid increases in proficiency might be expected as drills are learned. Thus, models 2–4 should be appropriate for predicting changes in proficiency for $\geq 16$ hours of training.

Model *3* included an intercept term, $\beta_0$, a linear term, $\beta_1$, describing the relationship between training time and proficiency, and a quadratic term, $\beta_2$, accounting for a possible threshold point after which time spent training begins to have either an increased or diminished effect:

$$\mu_3 = \beta_0 + \beta_1 x_{i\Delta t} + \beta_2 x_{i\Delta t}^2.$$

The prediction was that $\hat{\beta}_1 > 0$ (where hats ^ denote estimates), indicating an increase in skill improvement with increased training time; and $\hat{\beta}_2 < 0$, indicative of greater difficulty in increasing proficiency as training time and proficiency increase.

Finally, another general model, *4*, included a linear term describing the relationship between training time and performance, as well as an interaction term expressing the additional influence of starting proficiency level:

$$\mu_4 = \beta_0 + \beta_1 x_{i\Delta t} + \beta_3 y_{it_1} * x_{i\Delta t}.$$

The above model simply states that the average skill improvement can be written as a linear function of some overall baseline effect, $\beta_0$, an effect of additional time spent training, $\beta_1$, and an interaction effect between initial skill level and time spent training, $\beta_4$. The expectation was that $\hat{\beta}_1 > 0$, indicating an increase in skill improvement with increased training time; and $\hat{\beta}_3 < 0$, indicating a reduced effect of training for athletes with high initial skill levels. Note that unlike models *1–3*, each of which postulates a single relationship for all individuals in the population, model *4* produces a different relationship for individuals with different values for $y_{it_1}$, yielding a family of relationships associated with the different starting proficiencies.

Each of these 4 models was fit to the data, and model selection statistics and parameter estimates were then used to judge which model "best" described the data. Models were fit using maximum likelihood, and Akaike's Information Criterion (AIC) [14, 15] was used for model selection. Based on the assumed Gaussian model, the maximum likelihood estimator for each of the models *1–4* takes the form of the regression equation

$$\hat{\beta} = \min_\beta \|(y_i - \mu_\nu(\beta))^2\|, \ \nu = 1, 2, 3, 4. \tag{1}$$

The resulting parameter estimates provide information about the importance of each independent variable. Each effect estimate, $\hat{\beta}_l$, was examined to determine whether its sampling distribution, the width of which is reflected by the associated standard error (see **S1 Appendix**), $\hat{SE}(\hat{\beta}_i)$, was centered near 0 (indicating little support for the effect) or far from 0 (supporting the effect).

Because all 4 models represented plausible hypotheses, and because no single model received overwhelming support relative to the other models, model-averaged estimates [15, 16] were computed for the $\beta$ parameters that defined our different models (see **S1 Appendix**).

**Commitment.** It has been suggested that perhaps a better predictor of performance improvement is "grit" (popularized in [12]), a term intended to reflect not just the time spent training for an activity, but the quality of that training and the dedication of the individual athlete to it. This grit characteristic is referred to here as commitment and is quantified over the 10-week training period as described above. As with training time, an increase in improvement was predicted for those athletes with higher commitment scores. A stronger relationship was predicted for commitment than for training time, but because the scales of these 2 independent variables are not comparable, this expectation does not lead to predictions about relative magnitudes of the effect estimates, $\hat{\beta}_1$. Both training time and commitment could not be included as independent variables in the same model(s), because training time is a component of the commitment measure and these variables were relatively highly correlated. In order to avoid potentially problematic issues with collinearity, model selection was used to compare the models developed for these 2 independent variables.

The basic modeling approach described above for time spent training was also used for the commitment variable, $z_{it}$.

$z_{it}$ = training commitment accumulated by athlete $i$ through $\Delta t$ weeks of training.

The commitment variable corresponding to the end of the 10-week assessment period was denoted as $z_{i\Delta t}$ for consistency with the time variable. The same basic set of models as used for time spent training was used to assess the relationship between accumulated commitment and improvement in proficiency. Thus, $z_{i\Delta t}$ was substituted for $x_{i\Delta t}$ in all of the above expressions in order to conduct this second set of analyses.

**Training time vs. commitment.** In addition to the separate model sets for training time and commitment, all models were combined to create a model set including models for each variable as well as a single null model (no covariate effect). AIC scores and weights were then used to assess the relative abilities of these combined models to parsimoniously describe the processes generating the data. Evidence ratios were computed for specific pairs of models as $w_i/w_j$, where $w_i$ and $w_j$ denote the AIC weights (**S1 Appendix**) of the 2 models being compared. The evidence ratio reflects the degree to which evidence for model $i$ is greater than that for model $j$.

## Results

Scores were recorded for 108 individuals over a 10-week period in the spring of 2020. The data consist of the proficiency levels recorded at the beginning and end of the period of performance for each athlete. Data were also reported during the interim weeks; the number of reported scores help comprise the independent "commitment" variable. Also recorded are the accumulated hours trained over this period of performance. Initially, separate model sets are presented for each of the 2 independent variables, focusing on the nature of these relationships. The model selection statistics for the 2 independent variables are then combined in order to assess which variable provided the better descriptions of the data.

### Training time

Model selection statistics for time spent training are presented in Table 1. The "best" model ($\Delta$AIC = 0) was model 2, in which there was a linear increase in proficiency with training time. Model 3 was very competitive, showing evidence of an increase in proficiency with training time until about 80 hours, after which additional training did not appear to further increase proficiency (Fig 1). Model 4 received less support than models 2 and 3, but provided some evidence of greater increases in proficiency for athletes beginning training at lower proficiency levels. The lowest ranked model was model 1 (no improvement with training time). Estimated $\beta$ parameters describing these relationships (Table 1) provided strong evidence of an increase in proficiency with increased hours of training time.

**Table 1. Model parameter estimates and standard errors, along with the associated ΔAIC values indicating the relative support for each of the four models using "training time" as the independent variable.**

| Model | $\hat{\beta}_0(\hat{\sigma}_0)$ | $\hat{\beta}_1(\hat{\sigma}_1)$ | $\hat{\beta}_2(\hat{\sigma}_2)$ | $\hat{\beta}_3(\hat{\sigma}_3)$ | ΔAIC | $w_i$ |
|---|---|---|---|---|---|---|
| 2 | 0.6336(0.0965) | 0.0104(0.0018) | - | - | 0.0000 | 0.4659 |
| 3 | 0.2551(0.2156) | 0.0252(0.0078) | -0.0001(0.0001) | - | 0.0240 | 0.4603 |
| 4 | 0.6264(0.0972) | 0.0117(0.0025) | - | -0.0011(0.0014) | 3.6946 | 0.0735 |
| 1 | 1.1577(0.0422) | - | - | - | 14.6612 | 0.0003 |

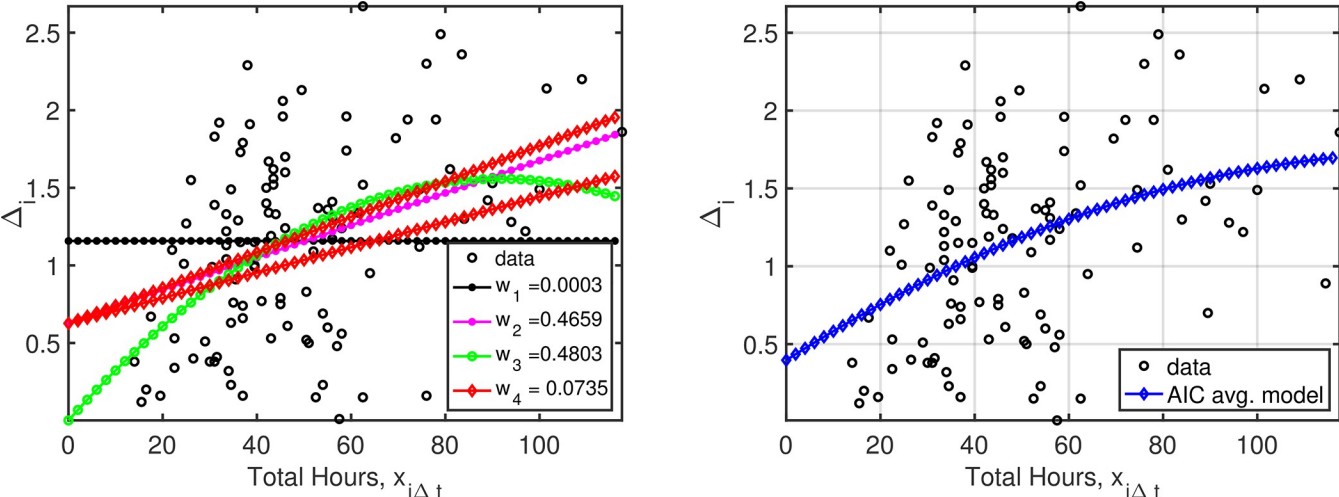

**Fig 1.** (left) Data and corresponding model weights for each of the 4 models described in the text. The model *4* plots are of the extrema coinciding with the smallest and largest interaction effects (the lowest and highest beginning skill levels), so that there are 2 curves associated with this model. (right) Weighted average model results obtained using AIC weights. All models predict improvement with training, however the exact nature of the relationship between improvement and training time is less clear. The model-averaged predictions (right) represent our best assessment of overall training effects, given this model uncertainty.

The raw data for each athlete in the training program, as well as the relationships between proficiency and training under all 4 models, are plotted in Fig 1. The raw data show substantial variation, emphasizing the need for statistical inference in order to draw conclusions about the effectiveness of training. Also shown in Fig 1 is the model-averaged relationship, which can be viewed as providing the best assessment of overall training effects, given model uncertainty. The model-averaged relationship shows increases in proficiency with hours of training, consistent with *a priori* hypotheses and the purpose of the training program.

## Commitment

Following the same modeling and estimation approach as above, the analyses were repeated with "commitment" score as the independent variable. The quadratic (model *3*) and linear (model *2*) models were both competitive, showing increases in skill improvement with larger commitment scores (Table 2, Fig 2). Given model uncertainty, the model-averaged plot in Fig 2 provides the best description of this relationship.

## Combined analysis for training time and commitment

To address the question of which independent variable provided a better description of the data, the training time and commitment models were combined to form a single model set with new model weights. In this "joint" assessment, seven separate models are considered, and

**Table 2. Model parameter estimates and standard errors, along with the associated ΔAIC values when using commitment as the independent variable.**

| Model | $\hat{\beta}_0(\hat{\sigma}_0)$ | $\hat{\beta}_1(\hat{\sigma}_2)$ | $\hat{\beta}_2(\hat{\sigma}_2)$ | $\hat{\beta}_3(\hat{\sigma}_3)$ | ΔAIC | $w_i$ |
|---|---|---|---|---|---|---|
| 3 | -0.9281(0.4387) | 2.9192(0.7880) | -0.8003(0.3319) | - | 0.0000 | 0.5800 |
| 2 | 0.0839(0.1287) | 1.0409(0.1197) | - | - | 0.9494 | 0.3608 |
| 4 | 0.0829(0.1291) | 0.9775(0.1407) | - | 0.0599(0.0695) | 4.5673 | 0.0591 |
| 1 | 1.1577(0.0422) | - | - | - | 31.9137 | 0.0000 |

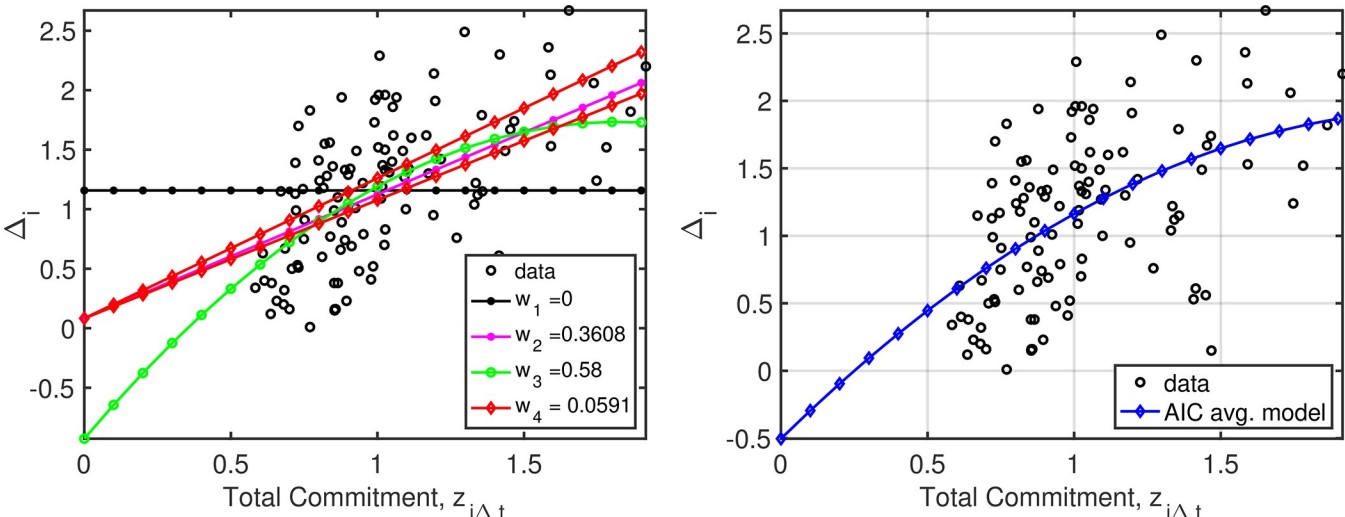

**Fig 2.** (left) Data and corresponding model weights for each of the 4 models described in the text, but with z (commitment) as the independent variable. For model 4, the extrema coinciding with the smallest and largest interaction effects are again plotted. (right) Weighted average model results obtained using AIC weights. The model averaged predictions (right) again represent the best assessment of overall training effects given the model set considered.

their AIC values are labeled as $AIC_{CT1}$, $AIC_{C2}$, $AIC_{C3}$, $AIC_{C4}$, $AIC_{T2}$, $AIC_{T3}$, $AIC_{T4}$, where the subscripts denote the independent variable (T = time, C = commitment), and the form of the model (1–4) for that independent variable. The first model (constant) is identical for both independent variables (time, commitment); hence only the single $AIC_{CT1}$ value need be considered.

The model weights corresponding to use of commitment as the independent variable are all considerably larger than those using training time as the independent variable (Table 3). This suggests that commitment is a better predictor of performance improvement than time spent training. The largest model weight corresponds to the model that predicts a quadratic relationship, showing an increase in performance with commitment score that levels off at higher levels of commitment. An evidence ratio was computed using the AIC weights of the best models for each covariate, models C3 and T2. The evidence ratio, $w_{G3}/w_{T2} = 5799$, strongly supported the quadratic commitment model as a better model for performance improvement than the linear training time model.

**Table 3. AIC, ΔAIC, and model weights obtained by considering the joint model set consisting of 6 models associated with using commitment (C2-C4) and training time (T2-T4) as the independent variables, as well as a constant (null) model, CT1.** The "constant" model is the same for the 2 independent variables.

| Model | AIC | ΔAIC | $w_i$ |
|---|---|---|---|
| C3 | 172.3841 | 0.0000 | 0.5799 |
| C2 | 173.3335 | 0.9494 | 0.3608 |
| C4 | 176.9514 | 4.5673 | 0.0591 |
| T2 | 189.6368 | 17.2527 | 0.0001 |
| T3 | 189.6608 | 17.2767 | 0.0001 |
| T4 | 193.3314 | 20.9473 | 0.0000 |
| CT1 | 204.2978 | 31.9137 | 0.0000 |

## Discussion

The general objective of this study was to evaluate whether participation in an athletic training program can increase proficiency in targeted athletic skills. The specific study objective focused on 2 metrics reflecting the degree of participation in the training: (1) hours spent in the program and (2) a measure of athlete commitment to the training. It was hypothesized that larger values of both participation metrics would lead to larger gains in proficiency, and that commitment might be the better of the 2 predictors of increased proficiency. Results provided strong evidence that both hours spent training and commitment score were positively associated with increased proficiency in the targeted skills. In addition, evidence strongly supported the hypothesis that commitment score was the better of the 2 predictors of increased proficiency.

The development of training programs designed to improve athletic performance is likely as old as competition itself. Athletes and trainers largely base training recommendations on their preconceptions and their personal experiences with training that has and has not appeared to "work", either for themselves or athletes whom they have trained. Learning from experience is certainly sensible, but humans frequently fail to do this in an objective manner. For example, humans tend to focus on observations that support their prior beliefs, subconsciously ignoring and downplaying observations that fail to support those beliefs. This tendency has been labeled "confirmation bias" and is well-known to psychologists [17], athletic trainers [3], and songwriters: "But a man hears what he wants to hear and disregards the rest," [18].

The training program investigated in this paper was developed by one of the authors (M. K.) who has many years of experience coaching and training young soccer players. The drills that comprised the tested training were all expected to lead to increases in proficiency of focal skills. However, the scatterplot of points resulting from this investigation showed substantial variation, illustrating the difficulties in drawing inferences based on data for human performance. For example, despite the majority of individuals who showed increases in proficiency with training, a few individuals showed negligible increases despite nearly 60 hours of training.

One possible explanation for such outliers is that the quality of the training time for some athletes was sufficiently low to prevent substantial improvement. It was hypothesized that perhaps use of "commitment" as the independent variable would be one way to assess both the amount and quality of the training and hence provide a more reliable predictor of success. Indeed, the aforementioned outliers scored much lower in terms of "commitment" relative to the population than they did for "hours trained". The joint model set including all models provided strong evidence that the relationship between commitment and performance improvement was much stronger than that obtained via training time alone (Table 3).

Methodologically, the selected approach to dealing with this variation and uncertainty was to use multimodel inference [15]. This approach deals not only with variation in observations conditional on a particular model being a good approximation to reality (e.g., the spread of points that do not fall exactly on a regression line), but also with the fact that humans never "know" which of several candidate models provides the best description of the processes that generated the data.

Based on this approach of multimodel inference, the best estimates of the relationship between hours expended training and increased proficiency are provided by model averaging and indicate roughly a 0.15-point increase in proficiency with every 10 hours of training, with smaller increases at higher training times. Given that the average of the initial skill scores is 1 point, this model can be viewed as predicting a 15% increase in proficiency per additional 10 hours trained, but with those increases beginning to level off after about 50 hours of training.

The relationship between commitment and skill improvement was best described by quadratic (C3) and linear (C2) models. The plot of model-averaged values provides the best estimate of this relationship, showing larger gains in proficiency for athletes with higher commitment scores.

Thus, the separate analyses with the 2 independent variables provided evidence of the importance of both training time and commitment to proficiency gains, as predicted. Model selection statistics based on the joint model set (Table 3) show that evidence is stronger for commitment. Results thus support the contention that both training time and individual commitment are key determinants of proficiency gains, with the addition to training time of individual characteristics related to the quality of training (commitment) increasing the explanatory ability of the models.

The basic approach used here can be used to answer a host of additional questions relating training to outcomes. One could, for example, relate the fraction of time spent on a particular skill in practice to improvement in game situations. Alternatively, one could expose multiple teams to different training regimens (e.g., vary ratio of speed and technical training) and compare the resulting values for $\beta_1$. A number of outstanding questions [3] relating training to proficiency, and proficiency to game impact, can be addressed using this approach.

This study was motivated by the fact that much of the conventional wisdom in the fields of athletic training and performance is based largely on personal experience. This approach has served athletics reasonably well, but suffers from the possibility that some ideas may attain acceptance simply because they are championed by vocal persons with strong personalities. More and more disciplines are focusing on evidence-based approaches to develop best practices. Statistical approaches such as those used here provide a means of deriving strong inferences using data from existing training programs. Even stronger inferences can be obtained by accumulating evidence of multiple analyses such as those presented here [19] and by use of true experimental approaches. Such efforts to obtain stronger inferences are recommended especially for questions characterized by substantial existing uncertainty.

In conclusion, this study focused on a training program developed to increase proficiency in specific soccer skills. Both number of hours devoted to this training and a measure of commitment to the training program were positively related to increases in proficiency, and evidence was particularly strong for commitment. Although the specific results of this study are limited to one particular training program, they are consistent with general hypotheses about the importance to athletic training of individual characteristics variously described as grit and commitment (12). To the extent that commitment can be encouraged and developed by coaches, this evidence of its importance has the potential to be very useful. Specifically, coaches and trainers can emphasize to athletes the importance of not just allocating time to training exercises, but also focusing on the quality of the training. However, even if commitment is entirely intrinsic to each athlete and cannot be developed, the gains in proficiency associated with training time alone attest to the utility of this training program. In addition to these specific applications of study results, the more general application of methods similar to those used in this study should lead to a stronger evidentiary basis for athletic training.

## Supporting information

**S1 Appendix. Akaike's Information Criterion (AIC), Akaike weights and multimodel inference.**
(DOCX)

**S1 Data.**
(DAT)

**S2 Data.**
(DAT)

## Author Contributions

**Conceptualization:** Cam M. K. Rechenmacher, Michael Keating, James D. Nichols, Jonathan M. Nichols.

**Data curation:** Cam M. K. Rechenmacher, Michael Keating.

**Formal analysis:** James D. Nichols, Jonathan M. Nichols.

**Investigation:** Michael Keating, Jonathan M. Nichols.

**Methodology:** James D. Nichols, Jonathan M. Nichols.

**Project administration:** Michael Keating.

**Resources:** Michael Keating.

**Software:** Jonathan M. Nichols.

**Supervision:** Michael Keating, James D. Nichols.

**Validation:** Cam M. K. Rechenmacher, Michael Keating, James D. Nichols, Jonathan M. Nichols.

**Writing – original draft:** Cam M. K. Rechenmacher, Michael Keating, James D. Nichols, Jonathan M. Nichols.

**Writing – review & editing:** Cam M. K. Rechenmacher, Michael Keating, James D. Nichols, Jonathan M. Nichols.

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
