## [Decision Letter · Decision Letter 0]

29 Jun 2022

PONE-D-22-04804An evidence-based approach to assessing the effectiveness of training regimen on athlete performance: youth soccer as a case studyPLOS ONE

Dear Dr. Nichols,

Thank you for submitting your manuscript to PLOS ONE. After careful consideration, we feel that it has merit but does not fully meet PLOS ONE’s publication criteria as it currently stands. Therefore, we invite you to submit a revised version of the manuscript that addresses the points raised during the review process. Considering the reviewers comments and requirement, it is suggested to make a major revision of your manuscript. 

We look forward to receiving your revised manuscript.

Kind regards,

Giancarlo Condello, Ph.D.

Academic Editor

PLOS ONE

“Unfunded study”

“No authors have competing interests”

Reviewers' comments:

Reviewer's Responses to Questions

**Comments to the Author**

1. Is the manuscript technically sound, and do the data support the conclusions?

Reviewer #1: Yes

Reviewer #2: Yes

2. Has the statistical analysis been performed appropriately and rigorously? 

Reviewer #1: I Don't Know

Reviewer #2: I Don't Know

3. Have the authors made all data underlying the findings in their manuscript fully available?

Reviewer #1: Yes

Reviewer #2: Yes

4. Is the manuscript presented in an intelligible fashion and written in standard English?

Reviewer #1: Yes

Reviewer #2: Yes

5. Review Comments to the Author

Reviewer #1: Dear Authors,

I appreciate the opportunity to comment to the authors in their manuscript titled "An evidence-based approach to assessing the effectiveness of training regimen on athlete performance: youth soccer as a case study". The manuscript's issue is interesting and relevant in the current context of youth soccer training knowledge. I consider that the relevance of the study lies in the limited research available exploring the formal evidentiary-basis for athletic training programs of academy soccer players.

The purpose of the study is based on the hypothesis that both increased training time and greater commitment would produce larger increases in performance improvement, and that commitment would be the most important determinant of improvement. The authors conclude with strong evidence for an increase in performance improvement with both training hours and commitment score.

The study is well structured, well written and has an innovative approach, which is why I believe it is suitable for publication and so I have noted this to the Editorial Board

Congratulations on the good work done.

BW

João Paulo Brito

Reviewer #2: The present study is of interest to investigate the effectiveness of training regimen on athlete performance, using a multi-model approach to estimate the relationship between skill improvement, hours and commitment.

Despite the interesting work, I strongly suggest following the comments to improve the quality of the manuscript.

Abstract

1. Authors should add a conclusion section.

Introduction

2. Line 35-37. "Coaches, trainers and athletes have searched for the optimal formula for skill acquisition for as long as sports have existed, simply because athletes who can develop abilities faster, or to a greater extent, than others possess a competitive advantage."

Authors should add a valid reference.

3. Line 37-38. "Historically, the selection of training methods has been guided by anecdotal experience rather than by empirical evidence on effectiveness."

Authors should add a valid reference.

4. Line 40-42. "However, from our varied experiences with elite club, high school, Division 1 collegiate, and professional athletics, we see that athletes and their mentors do not use empirical evidence to guide their training decisions as frequently as might be hoped."

With all due respect, I ask the authors how we can follow this rationale.

5. Line 44-45. "In some cases, opinions of coaches and trainers can seem to converge on a conventional wisdom."

What or wich cases? Be more specific using more details.

6. Line 78. "...and we focus on a 10-week period."

Please explain why, using valid scientific references.

7. Line 80. In my point of view, authors should avoid in scientific papers, using words such "etc". Please, mention all variables needed.

8. Line 85-86. "The basic hypothesis for the training time analysis was that improvement would be greater for athletes who expended more time training."

Please, explain why.

9. Line 90. "...aged 9-18 years who compete at varying levels (recreational through elite)."

How authors really classified the competitive and/or skill levels of the players? Please sustain your answer based on scientific evidence.

10. Line 92-93. "We deal with this variation in part by focusing on improvement in skill over the period of training, rather than on absolute skill level attained. " Please be more specific.

11. Line 95-96. "The basic idea was that athletes starting out at lower skill levels can increase proficiency rapidly. "

How this was really ensured?

12. Line 101-102. "...and was measured by accumulated activities."

Please, explain why.

Methods

13. Line 113. " ...we had NA=108 athletes..."

Please, explain why adding how athletes were really selected.

14. Line 117. "Ethics. We did not seek approval from an ethics committee because:"

Authors should confirm and add the approval/consent by the PlosOne officer if the paper could be published (if accepted) without ethical consent.

15. Line 127. "Consent was informed and documented via signature."

Authors should add, as a supplement file, one example of those informed consents.

16. Line 136-137. "This training requires periodic assessment tests to evaluate the rate of development of the athletes."

Please mention what type of test the authors are referring to.

17. Line 152-153. "Overall commitment scores were computed by dividing the number of points accrued by the maximum number of points for a player completing 100% of required tasks."

Authors should give more details/explanations how really this "equation" was created and calculated.

18.Line 153-154. "Optional tasks were available as well, such that commitment scores could range from 0 to 2.40 for each player."

Please, explain why.

19. Line 161. "Skill improvement" section.

Authors should add what type of skills were performed by the athletes. Were all the same for each age (9 to 18 years)?

20. Line 171-179. "Players measured their skill scores in the dribbling track based on how long it took them to perform each of the three designated drills. For the first touch and passing track, they recorded how many correct repetitions they could complete in 30 seconds, and for the other two drills they recorded how many consecutive, correctly-executed repetitions they could achieve before making a mistake (i.e., the ball drops). For the striking track, players measured all three of the drills as the distance from the goal that they could correctly complete each of the striking techniques, judging each striking drill separately as an average of the maximum left-footed and right footed distances they achieved. A new distance for either foot on any one of the drills could be achieved only by completing five correct strikes in a row with that technique."

All the previous information (i.e, each sentence), needs a valid references to better support all your rationale. Please, explain each sentence using valid references to better support how your skills were really selected.

21. Line 181. "...we developed a “Skill Stage” scoring system reflecting objective standards of skill achievement..."

I honestly ask if the authors already validated the mentioned score? If yes, please add the respective reference.

I truly recommend the authors to better explain how your main outcomes were selected and calculated from the online platform used (e.g, commitment and skill improvement). As it stands, in my point if view, the actual meaninglessness of your results can also be questioned. Moreover, more details are needed regarding how your variables were really treated and analysed before created a data set.

6. PLOS authors have the option to publish the peer review history of their article (what does this mean?). If published, this will include your full peer review and any attached files.

Reviewer #1: **Yes: **João Paulo Brito

Reviewer #2: **Yes: **Júlio Alejandro Henriques da Costa

---

## [Author Response · Author response to Decision Letter 0]

2 Aug 2022

Aug. 1, 2022

Giancarlo Condello, Ph.D.

Academic Editor

PLOS ONE

Re: PONE-D-22-04804

An evidence-based approach to assessing the effectiveness of training regimen on athlete performance: youth soccer as a case study

Dear Dr. Condello:

Thank you for your invitation to submit a revised version of our manuscript. As requested in your message of June 29, 2022, in this letter we respond to queries in your email and also detail our responses to the reviewers of this manuscript. We begin by copying your queries and responding in italics.

PLOS ONE requirements:

Response: We have reviewed the templates and believe that we have complied with style requirements.

“Unfunded study”

a)Please clarify the sources of funding (financial or material support) for your study. List the grants or organizations that supported your study, including funding received from your institution.

Response: No funding was received for this work from any source.

Response: There were no funders.

Response: There were no funders, and no author received any salary for this work.

Response: The authors received no specific funding for this work.

“No authors have competing interests”

Response: The authors have declared that no competing interests exist.

Responses to Reviewers’ Questions:

We simply copied the review comments and then placed our responses in italics below them. We very much appreciated the comments of reviewer 1. We tried to follow the recommendations of reviewer 2 as well. We appreciate the time and effort expended by you and both reviewers on this manuscript. 

Comments to the Author

1. Is the manuscript technically sound, and do the data support the conclusions?

Reviewer #1: Yes

Reviewer #2: Yes

2. Has the statistical analysis been performed appropriately and rigorously? 

Reviewer #1: I Don't Know

Reviewer #2: I Don't Know

3. Have the authors made all data underlying the findings in their manuscript fully available?

Reviewer #1: Yes

Reviewer #2: Yes

4. Is the manuscript presented in an intelligible fashion and written in standard English?

Reviewer #1: Yes

Reviewer #2: Yes

5. Review Comments to the Author

Reviewer #1: Dear Authors,

I appreciate the opportunity to comment to the authors in their manuscript titled "An evidence-based approach to assessing the effectiveness of training regimen on athlete performance: youth soccer as a case study". The manuscript's issue is interesting and relevant in the current context of youth soccer training knowledge. I consider that the relevance of the study lies in the limited research available exploring the formal evidentiary-basis for athletic training programs of academy soccer players.

The purpose of the study is based on the hypothesis that both increased training time and greater commitment would produce larger increases in performance improvement, and that commitment would be the most important determinant of improvement. The authors conclude with strong evidence for an increase in performance improvement with both training hours and commitment score.

The study is well structured, well written and has an innovative approach, which is why I believe it is suitable for publication and so I have noted this to the Editorial Board

Congratulations on the good work done.

BW

João Paulo Brito

Response: We very much appreciate the comments and support of our work by reviewer 1. 

Reviewer #2: The present study is of interest to investigate the effectiveness of training regimen on athlete performance, using a multi-model approach to estimate the relationship between skill improvement, hours and commitment.

Despite the interesting work, I strongly suggest following the comments to improve the quality of the manuscript.

Abstract

1. Authors should add a conclusion section.

Response: The abstract concludes with the following 3 sentences: 

“Despite considerable variability in the data, we find strong evidence for an increase in performance improvement with both training hours and commitment score. We compared the best models for hours and commitment by computing an evidence ratio of 5799, indicating much stronger evidence favoring the model based on commitment. Results of analyses such as these go beyond anecdotal experience in an effort to establish a formal evidentiary basis for athletic training programs.”

We view these as the central conclusions of the paper, however we are happy to provide specific additions or clarifications if requested. 

Introduction

2. Line 35-37. "Coaches, trainers and athletes have searched for the optimal formula for skill acquisition for as long as sports have existed, simply because athletes who can develop abilities faster, or to a greater extent, than others possess a competitive advantage."

Authors should add a valid reference.

Response: We have added 2 relevant citations as the reviewer recommended. 

3. Line 37-38. "Historically, the selection of training methods has been guided by anecdotal experience rather than by empirical evidence on effectiveness."

Authors should add a valid reference.

Response: As recommended by the reviewer, we added 2 references to complement our personal observations.

4. Line 40-42. "However, from our varied experiences with elite club, high school, Division 1 collegiate, and professional athletics, we see that athletes and their mentors do not use empirical evidence to guide their training decisions as frequently as might be hoped."

With all due respect, I ask the authors how we can follow this rationale.

Response: We were simply trying to convey that we have often observed failures to use empirical evidence to guide training at every level of sport. This observation is consistent with the references that we have added in response to the reviewer’s previous recommendation. We believe these added citations properly address the reviewers’ concerns. 

5. Line 44-45. "In some cases, opinions of coaches and trainers can seem to converge on a conventional wisdom."

What or wich cases? Be more specific using more details.

Response: The sentence that follows lines 44-45 (“However, the recent revolution in analytics provides strong evidence of the fallibility of such wisdom”) contains references to 2 entire books that are devoted to examples of both long-held conventional wisdom and empirical evidence (provided by the recent interest in sports “analytics”) overturning such wisdom. One example from reference 6 derives from professional baseball, where a hitter’s worth was judged by a few primary statistics that included “batting average”, the fraction of “at bats” at which a hitter got a base hit. This reliance on batting average was virtually universal across all levels of baseball play (amateur to professional). Analytics showed that ‘worth’ with respect to a team’s ability to win is more strongly predicted by a hitter’s “on base percentage”, a statistic that had been computed, but was little used to assess value or worth of a player. There are many such examples from many sports, and we believe the 2 cited books are sufficient to reinforce this point. Nonetheless, if the editor prefers, we are happy to provide material providing additional examples. 

6. Line 78. "...and we focus on a 10-week period."

Please explain why, using valid scientific references.

Response: 10 weeks was the standard time of each training course, and was thus a natural duration for data collection. We specified this in the revised manuscript as recommended by the reviewer. We also cited a paper that was the basis for our selection of 10 weeks. This paper recommended 66 days as the time required for humans to develop new habits. 

7. Line 80. In my point of view, authors should avoid in scientific papers, using words such "etc". Please, mention all variables needed.

Response: We omitted “etc.” as recommended by the reviewer, but chose not to attempt to enumerate every single factor that might influence how an athlete might choose to spend her time. Instead, we provided 2 examples of relevant factors. 

8. Line 85-86. "The basic hypothesis for the training time analysis was that improvement would be greater for athletes who expended more time training."

Please, explain why.

Response: The importance of training and practice to achievement in any endeavor is encoded in conventional wisdom (e.g., sayings such as “practice makes perfect”) and is a belief widely shared by athletes and coaches alike. Nonetheless, we have added 2 citations (citations 2 and 11) that explain and provide evidence for the importance of training. Both references are from K.A. Ericsson, an authority on the duration and quality of practice as it relates to skill improvement. 

9. Line 90. "...aged 9-18 years who compete at varying levels (recreational through elite)."

How authors really classified the competitive and/or skill levels of the players? Please sustain your answer based on scientific evidence.

Response: A useful aspect of our analysis is that it does not require classification of athletes by initial skill levels. Because we focus on the precise difference in test scores before and after training, we explicitly account for the starting (pre-training) ability level of each athlete, without having to classify them (see line 93-95). We also investigated one model that permitted an effect of before-training test scores on the rate of increase in ability with training. In short, our hypotheses and subsequent analyses permitted us to use the test scores directly without the need to place athletes into classes (see our response to comment 10 below for additional details). 

10. Line 92-93. "We deal with this variation in part by focusing on improvement in skill over the period of training, rather than on absolute skill level attained. " Please be more specific.

Response: We added a note to “see Analytics Methods”, as the exact expression to compute the difference is presented there (page 10, line 209 – see also inclusion of starting proficiency in model 4, line 246 page 12)

11. Line 95-96. "The basic idea was that athletes starting out at lower skill levels can increase proficiency rapidly."

How this was really ensured?

Response: The hypothesis that athletes beginning at lower skill levels might increase proficiency more rapidly than athletes starting at higher levels is not ensured at all. Rather this was a hypothesis to be tested, so we incorporated this hypothesis into one of our models (model 4, line 246) and not into the others. We thus tested this idea and found little support for it (see e.g., table 1 or table 3). 

12. Line 101-102. "...and was measured by accumulated activities."

Please, explain why.

Response: We wanted our measure of commitment to reflect activities occurring over the duration of the training period, just as training time was measured over the entire 10-week program. We added a parenthetical note to “see Training Methods and Metrics”, as details of our approach are described in that section. 

Methods

13. Line 113. " ...we had NA=108 athletes..."

Please, explain why adding how athletes were really selected.

Response: Previously (lines 91-92), we wrote: “For this analysis we selected a relatively homogeneous group of athletes, focusing on soccer players aged 9-18 years who compete at varying levels (recreational through elite).” We specified that we selected athletes with “usable data from the Spring 2020 training program” (page 6, line 116). This specification of “usable data” simply meant that we did not include data from individuals who did not complete the program for any reason (e.g., injury), and we have added a statement to this effect in the manuscript (lines 116-117).

14. Line 117. "Ethics. We did not seek approval from an ethics committee because:"

Authors should confirm and add the approval/consent by the PlosOne officer if the paper could be published (if accepted) without ethical consent.

Response: We have described our consent process in accordance with the submission instructions, and PlosOne has found it adequate.

15. Line 127. "Consent was informed and documented via signature."

Authors should add, as a supplement file, one example of those informed consents.

Response: The exact consent statement is provided directly in the manuscript itself (lines 133-138).

16. Line 136-137. "This training requires periodic assessment tests to evaluate the rate of development of the athletes."

Please mention what type of test the authors are referring to.

Response: The testing simply entails repeating various soccer drills and videotaping them to ensure that reported scores are accurate. This general process and the exact drills are described in the text under Skill improvement, and we have therefore added a parenthetical note citing this section in response to the reviewer comment.

17. Line 152-153. "Overall commitment scores were computed by dividing the number of points accrued by the maximum number of points for a player completing 100% of required tasks."

Authors should give more details/explanations how really this "equation" was created and calculated.

Response: We have added several sentences of explanation as recommended by the reviewer (appearing on lines 156-165).

18.Line 153-154. "Optional tasks were available as well, such that commitment scores could range from 0 to 2.40 for each player."

Please, explain why.

Response: The completion of optional tasks (i.e., going above and beyond what was required) provided a direct indication of the commitment shown by athletes to the training program and self-improvement. Our rationale is explained in the following added statement (lines 162-165 ): “The commitment metric was intended to go beyond time expended on training, as it incorporated information that reflected an athlete’s commitment to completing all tasks required of them, and even some tasks that were not required.” 

19. Line 161. "Skill improvement" section.

Authors should add what type of skills were performed by the athletes. Were all the same for each age (9 to 18 years)?

Response: The skills and how they were measured are described in some detail in this section. We did not write of any age stratification because athletes of all ages worked on the same skills, albeit with different levels of proficiency. As a reminder, we dealt with this age (and other) variation in proficiency by focusing on improvement in proficiency, rather than absolute proficiency, in all analyses. 

20. Line 171-179. "Players measured their skill scores in the dribbling track based on how long it took them to perform each of the three designated drills. For the first touch and passing track, they recorded how many correct repetitions they could complete in 30 seconds, and for the other two drills they recorded how many consecutive, correctly-executed repetitions they could achieve before making a mistake (i.e., the ball drops). For the striking track, players measured all three of the drills as the distance from the goal that they could correctly complete each of the striking techniques, judging each striking drill separately as an average of the maximum left-footed and right footed distances they achieved. A new distance for either foot on any one of the drills could be achieved only by completing five correct strikes in a row with that technique."

All the previous information (i.e, each sentence), needs a valid references to better support all your rationale. Please, explain each sentence using valid references to better support how your skills were really selected.

Response: The reviewer recommends that we provide a reference for each statement describing our assessment drills. These drills were developed by author MK in order to assess skills that he thought to be important and sought to develop. These tests can be viewed as analogous to those designed by any university professor in order to assess learning in her or his specific class. We know of no standardized tests that are universally accepted in the soccer community, but even if such tests existed, it makes more sense to us to develop tests that are tailored to assessing the exact skills that we hoped to teach. As an aside, all coauthors have substantial experience with playing and coaching soccer, and all of us believe the drills to provide excellent assessments of proficiency in the selected skills. 

21. Line 181. "...we developed a “Skill Stage” scoring system reflecting objective standards of skill achievement..."

I honestly ask if the authors already validated the mentioned score? If yes, please add the respective reference.

Response: The reviewer asks us to “validate” our scores, but we are not certain exactly what this means. One possibility is that this means the reviewer would like it if someone else had used the exact same approach to address the questions that we address. But as noted above, our preference is to develop scoring systems that are tailored to our objectives. We believe this to be far preferable to borrowing a scoring system developed by someone else and hoping that it corresponded closely enough to our objectives. Once again, we return to the analogy of the university professor deciding how to weight and combine the different assessment instruments (tests, essays, class projects, etc.) of the semester in order to develop an assessment score that corresponds to her/his objectives. The other possible meaning of validation is that the reviewer would like us to provide our own evidence that the tests measure proficiency in the selected skills. Our results of higher test scores with more training and commitment actually provide such evidence. 

I truly recommend the authors to better explain how your main outcomes were selected and calculated from the online platform used (e.g, commitment and skill improvement). As it stands, in my point if view, the actual meaninglessness of your results can also be questioned. Moreover, more details are needed regarding how your variables were really treated and analysed before created a data set.

Response: This general comment about “how your main outcomes were selected and calculated” addresses 2 separate issues. (1) We agree with the reviewer that how the outcomes (test statistics) were calculated is very important. We have followed his suggestion about including additional information on our computation of a commitment score (see again the added paragraph now appearing on lines 156-165). In addition, we note that the explicit descriptions of how “variables were treated and really analysed” appear in the section Analytic methods. We believe that this section is very detailed, as it includes the explicit models fit to the data. But we can present even more detail if we know exactly what else is needed. (2) The second issue raised by the reviewer involves selection of test drills, namely that because we developed our own assessment drills (reviewer’s “main outcomes”) our results are somehow suspect. As noted above, we developed these drills specifically with our training program objectives in mind, and selected tests that we thought best evaluated whether our training program was successful in increasing proficiency in selected skills useful to soccer players. We have emphasized that this approach to testing is pervasive throughout academia, as well as the sports world. As a posteriori support that our drills and scoring system were appropriate for assessing our training program, we note that achievement based on this system showed improvement with increased hours of training and commitment to the program, as predicted. If we had done a poor job of assessment scoring, we would expect little evidence of a relationship between such scores and training/commitment. Finally, as we observed previously, we believe that an experienced soccer coach or player would consider these drills to provide very reasonable assessments of the specified soccer skills. 

We hope that the above responses seem reasonable to you. We made one additional change in the revised manuscript that was not in response to reviewer comments. The senior author was married recently, so we now use her new, married name. We look forward to hearing from you, and we will try to address any remaining concerns you may have. Thank you for considering our manuscript.

Sincerely,

James D. Nichols

---

## [Decision Letter · Decision Letter 1]

19 Sep 2022

PONE-D-22-04804R1An evidence-based approach to assessing the effectiveness of training regimen on athlete performance: youth soccer as a case studyPLOS ONE

Dear Dr. Nichols,

Thank you for submitting your manuscript to PLOS ONE. After careful consideration, we feel that it has merit but does not fully meet PLOS ONE’s publication criteria as it currently stands. Therefore, we invite you to submit a revised version of the manuscript that addresses the points raised during the review process.

ACADEMIC EDITOR:Dear authors,

Considering the work made by the previous editors and the revisions made by both reviewers, I also agree that this work had conditions to be published.

However, there minor details that can be improved. First an English revision/proofreading should be performed because there are several expressions and sentences not well written. Moreover, the work was written in the first person, but it should be changed to the third. Discussion should start with the aims of the study and the main results. In addition, there should be a limitation, a clear practical application and conclusion sections for this paper.

I believe these details will improve organization and clarity of the work.

Best regards

We look forward to receiving your revised manuscript.

Kind regards,

Rafael Franco Soares Oliveira

Academic Editor

PLOS ONE

Journal Requirements:

Additional Editor Comments:

Dear authors,

Considering the work made by the previous editors and the revisions made by both reviewers, I also agree that this work had conditions to be published.

However, there minor details that can be improved. First an English revision/proofreading should be performed because there are several expressions and sentences not well written. Moreover, the work was written in the first person, but it should be changed to the third. Discussion should start with the aims of the study and the main results. In addition, there should be a limitation, a clear practical application and conclusion sections for this paper.

I believe these details will improve organization and clarity of the work.

Best regards

Reviewers' comments:

Reviewer's Responses to Questions

**Comments to the Author**

1. If the authors have adequately addressed your comments raised in a previous round of review and you feel that this manuscript is now acceptable for publication, you may indicate that here to bypass the “Comments to the Author” section, enter your conflict of interest statement in the “Confidential to Editor” section, and submit your "Accept" recommendation.

Reviewer #1: All comments have been addressed

Reviewer #2: All comments have been addressed

2. Is the manuscript technically sound, and do the data support the conclusions?

Reviewer #1: Yes

Reviewer #2: Yes

3. Has the statistical analysis been performed appropriately and rigorously? 

Reviewer #1: I Don't Know

Reviewer #2: I Don't Know

4. Have the authors made all data underlying the findings in their manuscript fully available?

Reviewer #1: Yes

Reviewer #2: Yes

5. Is the manuscript presented in an intelligible fashion and written in standard English?

Reviewer #1: Yes

Reviewer #2: Yes

6. Review Comments to the Author

Reviewer #1: Dear Authors,

As I commented in the 1st review, I consider the manuscript to be suitable for publication and that was the note given to the editor.

Congratulations on the quality of the manuscript.

Reviewer #2: I am happy with the current version of the manuscript.

The authors did a good job on reviewing the manuscript and answering all the revisions maded.

7. PLOS authors have the option to publish the peer review history of their article (what does this mean?). If published, this will include your full peer review and any attached files.

Reviewer #1: **Yes: **João Paulo Brito

Reviewer #2: **Yes: **Júlio Alejandro Henriques da Costa

---

## [Author Response · Author response to Decision Letter 1]

7 Oct 2022

Dear Dr. Oliviera:

Thank you for your invitation to submit a revised version of our manuscript. As requested in your message of September 19, 2022, in this letter we respond to your comments and recommendations. Below, we will copy your main comments and respond in italics.

“First an English revision/proofreading should be performed because there are several expressions and sentences not well written. Moreover, the work was written in the first person, but it should be changed to the third.”

We have tried to improve and clarify the writing throughout, and we have rewritten the entire manuscript and appendix in the third person. As you can see in the Track Changes copy of our revision, virtually all paragraphs contain changes associated with these recommendations. 

“Discussion should start with the aims of the study and the main results. In addition, there should be a limitation, a clear practical application and conclusion sections for this paper.”

We have rewritten portions of the Discussion in order to follow your suggestions. We added a new paragraph to begin the Discussion (lines 664-672) in which we stated study aims and results. The primary limitation of the study is that inferences are based on a single training program (see lines 761-763). Practical applications are both specific (coaches/trainers encouraging greater commitment; lines 763-768) and general (use of similar methodological approaches to promote evidence-based training; lines 768-770; more detail and recommendations in lines 642-656). A conclusions paragraph was added (lines 725-757). 

We tried to follow all of your recommendations and hope that our responses seem reasonable to you. We look forward to hearing from you, and we will try to address any remaining concerns you may have. Thank you for considering our manuscript.

---

## [Editor Report · Decision Letter 2]

13 Oct 2022

An evidence-based approach to assessing the effectiveness of training regimen on athlete performance: youth soccer as a case study

PONE-D-22-04804R2

Dear Dr. Nichols,

We’re pleased to inform you that your manuscript has been judged scientifically suitable for publication and will be formally accepted for publication once it meets all outstanding technical requirements.

Kind regards,

Rafael Franco Soares Oliveira

Academic Editor

PLOS ONE

Additional Editor Comments (optional):

Dear authors,

Congratulations on the improvements made on your work. My recommendation is to accept your work for publication.

Best regards
---

## [Editor Report · Acceptance letter]

17 Oct 2022

PONE-D-22-04804R2 

An Evidence-Based Approach to Assessing the Effectiveness of Training Regimen on Athlete Performance: Youth Soccer as a Case Study 

Dear Dr. Nichols:

I'm pleased to inform you that your manuscript has been deemed suitable for publication in PLOS ONE. Congratulations! Your manuscript is now with our production department. 

Kind regards, 

on behalf of

Dr. Rafael Franco Soares Oliveira 

Academic Editor

PLOS ONE